# High-Performance n-Type Bi_2_Te_3_ Thermoelectric Fibers with Oriented Crystal Nanosheets

**DOI:** 10.3390/nano13020326

**Published:** 2023-01-12

**Authors:** Min Sun, Pengyu Zhang, Guowu Tang, Dongdan Chen, Qi Qian, Zhongmin Yang

**Affiliations:** 1State Key Laboratory of Luminescent Materials and Devices, Institute of Optical Communication Materials, Guangdong Provincial Key Laboratory of Fiber Laser Materials and Applied Techniques, and Guangdong Engineering Technology Research and Development Center of Special Optical Fiber Materials and Devices, School of Materials Science and Engineering, South China University of Technology, Guangzhou 510640, China; 2Nanjing Institute of Future Energy System, Nanjing 211135, China; 3School of Physics and Optoelectronic Engineering, Guangdong University of Technology, Guangzhou 510006, China

**Keywords:** n-type Bi_2_Te_3_, thermoelectric fibers, thermal drawing, crystalline orientation

## Abstract

High-performance thermoelectric fibers with n-type bismuth telluride (Bi_2_Te_3_) core were prepared by thermal drawing. The nanosheet microstructures of the Bi_2_Te_3_ core were tailored by the whole annealing and Bridgman annealing processes, respectively. The influence of the annealing processes on the microstructure and thermoelectric performance was investigated. As a result of the enhanced crystalline orientation of Bi_2_Te_3_ core caused by the above two kinds of annealing processes, both the electrical conductivity and thermal conductivity could be improved. Hence, the thermoelectric performance was enhanced, that is, the optimized dimensionless figure of merit (*ZT*) after the Bridgman annealing processes increased from 0.48 to about 1 at room temperature.

## 1. Introduction

Thermoelectrics (TEs) can directly convert heat to electricity through the directional movement of the internal carriers with the temperature difference. Due to their small size, high reliability, and no noise, thermoelectric devices have great potential for civil and military applications [1,2]. The dimensionless figure of merit (*ZT*) is used to identify the performance of TEs, which is determined by the inherent electrical conductivities (*σ*), Seebeck coefficients (*S*), power factors (*PF = S*^2^*σ*), and thermal conductivities (*κ*). Nowadays, bismuth telluride (Bi_2_Te_3_) is considered a class of state-of-the-art TEs (*ZT* > 1) for low-temperature applications (0–300 °C), such as Peltier refrigerators or CCD coolers [3]. Due to its anisotropic structure, Bi_2_Te_3_ is considered as a significant candidate for directional thermoelectric properties, such as the *c* plane or across the out-of-plane of Bi_2_Te_3_ sheets [4,5]. Theoretically, the ability of crystalline orientation is highly related to the crystal size and orientation degree of the *c* plane (*F*) [6]. There have been many studies on the introduction of crystalline orientation at various scales [7,8,9,10,11,12]; however, the artificial regulation of crystalline orientation and related mechanisms have rarely been reported.

In high-performance thermoelectric fiber devices, p-n Bi_2_Te_3_ fibers are usually paired in series electrically and in parallel thermally to realize thermoelectric conversion [13]. To date, it has been found that the n-type Bi_2_Te_3_ fibers (thermally drawn with glass cladding by a powder-in-tube method) could exhibit a restricted *ZT* < 0.5 owing to the low relative density and more texturing-related sensitivity of the carriers’ mobility than their p-type fibers (*ZT*~1.4) [9]. In the thermoelectric fibers fabricated by thermal drawing from our previous works [14,15,16,17], the *c*-plane crystalline orientation enhances the carrier mobility and electrical conductivity in Bi_2_Te_3_ core fibers.

Herein, high-performance n-type Bi_2_Te_3_ fibers were fabricated by a rod-in-tube thermal drawing method, and subsequent annealing processes (whole annealing and Bridgman annealing) were used to enhance the crystallization of two kinds of opposite crystalline orientations [17]. Crystalline orientations along (110) and (001) in the fiber cores were increased by whole annealing and Bridgman annealing, respectively, and the elemental enrichment was reduced. Hence, the resulting fibers exhibit tailored electrical-phonon transport, and the Bridgman-annealed fibers show a better thermoelectric performance. The optimized *ZT* values of the annealed fibers are measured to be about 1 near room temperature, which is about twice as large as that of their as-drawn fiber counterparts. The n-type Bi_2_Te_3_ fibers with oriented crystal nanosheets in the core and the glass cladding protection can be connected by electroconductive paste electrically in series and thermally in parallel with the similar p-type Bi_2_Te_3_ fibers to fabricate thermoelectric elements [6,9,13]. Additionally, they could be used in the field of waste heat recycling on the curved surface (e.g., hot water tubes and vehicle tailpipes), Peltier cooling, and temperature-sensing textiles (e.g., face masks and sleeveless shirts), etc.

## 2. Experimental Procedure

### 2.1. Fabrication

A two-step method with thermal drawing and post-processing annealing was applied on n-type Bi_2_Te_3_ fibers fabrication. The Bi_2_Te_3_ and Bi_2_Se_3_ powders of 99.999% purity (under 200 mesh, Aladdin, Ontario, CA, USA) at a ratio of 9:1 were used as raw materials to prepare an n-type Bi_2_Te_3_ rod with a relative density of 99%, and the fabrication process is reported in detail in our previous study [16]. The Bi_2_Te_3_ rod was inserted into a borosilicate glass (BK7, Schott, Zagreb, Croatia) tube, which has a glass-transition temperature of 562 °C and a softening temperature of 800 °C, forming a fiber preform. Several-meter-length Bi_2_Te_3_-core/glass-clad fibers were drawn from the fiber preform at ~880 °C in an optical fiber drawing tower under an argon atmosphere.

One group of the as-drawn fibers were wholly annealed at 565 °C for 5 h in a muffle furnace and cooled down to room temperature at a rate of 0.1 °C/min. The other group of the as-drawn fibers were Bridgman-method annealed, which descended step by step and crossed a high-temperature ring with a constant speed of 1 cm/h to recrystallize the core as shown in our previous study [6]. The ring-zone temperature is ~645 °C, which is higher than the melting temperature of the Bi_2_Te_3_ ~585 °C.

### 2.2. Measurements

For characterizing the crystallinity and electrical transport of the fiber cores, the as-drawn and annealed fibers were etched in HF acid to strip the glass cladding, and then, the fiber cores were identified by X-ray diffractometer (XRD, X’Pert PROX, Cu K*α*). Energy-dispersive spectroscopy (EDS) of elemental studies was carried out on the fiber cross-sections by using scanning electron microscopy (SEM, Zeiss Merlin, Oberkochen, Germany).

The Seebeck coefficients (*S*) and the electrical conductivities (*σ*) of the fiber cores were tested by a four-probe method [9]. Three-time measurements were performed under the same conditions for each fiber core to obtain the average testing value, and the relative deviations were <5%. The thermal conductivity (*κ*) was tested by a method of time-domain thermal reflection, and the relative deviation was <10%. All the relative deviations show the testing results are reproducible and reliable.

## 3. Results and Discussion

### 3.1. Microstructure

The XRD patterns of as-drawn fiber cores, whole-annealed fiber cores, and Bridgman-annealed fiber cores are exhibited in Figure 1. All the XRD peaks are indexed to the Bi_2_Te_2.7_Se_0.3_ hexagonal phase (JCPDS#50-0954). The as-drawn Bi_2_Te_2.7_Se_0.3_ fiber cores (BTSF) are polycrystals after a process of thermal drawing and quick cooling (>100 °C/s). The average crystal size of as-drawn fiber core is estimated to be ~30 nm based on the XRD peak width and the Scherrer formula [16], and the average particle sizes of post-annealed fiber cores are >100 nm. A great difference is found in XRD peaks among the as-drawn fibers and annealed fibers. The annealed fiber cores show larger XRD peak intensities than the as-drawn fiber cores at special lattice planes, such as (1 1 0) marked in blue or (0 0 6) marked in red. The difference in XRD peak intensities demonstrates that the crystals in annealed fiber cores could possess an orientation over the crystals in as-drawn fiber cores. According to the Lotgering method [6], the orientation degree *F* of the (0 0 1) planes in polycrystals can be calculated: *F* = (*P* − *P*_0_) / (1 − *P*_0_); *P*_0_ = *I*_0_(0 0 *l*) / ∑*I*_0_(*h k l*); *P* = *I*(0 0 *l*) / ∑*I*(*h k l*). Hence, the *F* of the crystals in as-drawn fiber cores and annealed fiber cores are calculated as 0.45, 0.06, and 0.92, respectively. This means that the as-drawn fiber cores and Bridgman-annealed fiber cores are in the (0 0 1) orientation and that the Bridgman-annealed fiber cores show a greater orientation. Oppositely, the whole-annealed fiber cores show a great (1 1 0) orientation when their (110) orientation factor (*F*_(110)_~0.85) can be calculated from the data of Figure 1.

The cross-section SEM images and EDS elemental mappings of Bi, Se, and Te on the as-drawn and annealed fibers are shown in Figure 2. All three samples show a layered structure in the cross-section. In Figure 2a,i, it is shown that the three samples show similar nanosheet microstructures, following the (001) orientation of the as-drawn/Bridgman-annealed fiber cores. It can be carefully observed that the nanosheets are almost parallel to the cross-section at the bottom left of the whole-annealed fiber core in Figure 2e, following the (110) orientation of the whole-annealed fiber cores. In the elemental mappings, there are traces of elements Bi, Se, and Te diffused from the core into the cladding region. It is observed that there are Te enrichments in the as-drawn fiber core, but little enrichment can be found in the annealed fiber cores. Enriched Te could be decreased by whole annealing at 565 °C or Bridgman annealing at 645 °C with a 1 cm/h recrystallization speed. Since the crystalline orientation and Te enrichments impact the electrical-phonon transport [18], the fiber with different Te enrichments may show diverse TE performance.

In the micrometer-scale limited space of fiber cores, it is important to uncover the growth process and mechanism of the oppositely crystalline orientation, whose nanosheet microstructure and element distribution are shown in Figure 2. As the as-drawn fibers went through a quick cooling process (>100 °C/s) after thermal drawing, as reported in our previous work [6], the fiber cores possessed a preferred (001) orientation because of residual thermal stress along the radial direction from the core to the cladding. The microstructure evolutions of the as-drawn fibers during annealing are schematically illustrated in Figure 3. In the whole annealing process (i), the thermal stress gradually decreased by annealing and slow cooling. The fiber core then exhibits a trend of being a preferred (110) orientation with the thermal stress decreasing. In the Bridgman annealing process (ii), the core melt gradually recrystallizes into nanosheets around the fiber interface to minimize surface energy. In the limited fiber core space, the nanosheets recrystallize continuously along the (001) plane and the fiber axis. It should be illuminated in a similar microstructural orientation during a single-crystal crystallization process by directional Bridgman annealing [19] or laser annealing [20]. In addition, there is elemental diffusion on the core–clad interface, which might cause surface roughness and low-dimensional defects of the fiber core, as in our previous study [9].

### 3.2. Thermoelectric Properties

The electrical conductivities (*σ*) and Seebeck coefficients (*S*) of the fibers are shown in Figure 4a,b, measured by a home-made setup in our previous work [6]. In Figure 4a, all the as-drawn/annealed fibers possess a decreasing *σ* with increasing temperature (10–50 °C), revealing the metallic resistance characteristic [21]. The *σ*_w_ of the whole-annealed fibers and the *σ*_b_ of the Bridgman-annealed fibers are higher than the *σ*_a_ of the as-drawn fibers, and the *σ*_b_ is approaching threefold *σ*_a_ at the same temperature. The result could be derived from the fact that the (001) orientation of Bridgman-annealed fibers or the (110) orientation of whole-annealed fibers could enhance *σ*, and the greater orientation along (001) would support the higher carrier mobility and *σ,* as reported in our previous work [6]. In Figure 4b, the negative Seebeck coefficient means that all the fibers are n-type conductors [22]. All the as-drawn fibers and the annealed fibers possess an increasing |*S*| with increasing temperature (10–50 °C). The |*S*_w_| of the whole-annealed fibers is bigger than |*S*_b_| of the Bridgman-annealed fibers and |*S*_a_| of the as-drawn fibers under the same temperature. This could be attributed to the fact that the annealing process could decrease Te enrichment to increase |*S*|, and the oriented crystals along (110) should exhibit a little higher |*S*| than the oriented crystals along (001), whose isotropic behavior is following the result of the reported n-type Bi_2_T_3_ single crystals [23] or films [24]. The *PF = S*^2^*σ* of all fibers were calculated and shown in Figure 4c. The *PF*_b_ of the Bridgman-annealed fibers is higher than the *PF*_w_ of the whole-annealed fibers and the *PF*_a_ of the as-drawn fibers at the same temperature. The *PF*_b_ of the Bridgman-annealed fibers exceeds fourfold the *PF*_a_, and the highest value is about 4 mW/mK^2^ at 10 °C.

The measured *σ*, *S*, *κ*, and calculated *ZT* of all three samples are listed in Table 1. For the thermal conductivities, the *κ*_a_ of the as-drawn fiber is ultralow, which could be caused by enhanced phonon scattering from the nanocrystalline grains, as reported in our previous work [16]. The room temperature *ZT* (~27 °C) of the Bridgman-annealed Bi_2_Te_2.7_Se_0.3_ fibers is the highest, which is twice as large as that of the as-drawn fibers and is two times larger than the reported *ZT* of the Bi_2_Te_2.5_Se_0.5_ or Bi_2_Se_3_ fibers [13,15]. Even the Bridgman-annealed fibers possess a higher *ZT* than that of the Bridgman-growth-method Bi-Te-Se crystals [25], which shows higher *σ* and *κ* but a lower *S* at the end of Table 1. For the high *ZT*, the Bridgman-annealed fibers possess a higher *σ* and a higher *κ* than Bi_2_Te_2.5_Se_0.5_ fibers, which benefits from the (001) orientation. Additionally, the whole-annealed fibers possess a higher *σ* but a similar *κ*, which also benefits from the (110) orientation. Different from the reported works found in oriented Bi_2_Te_3_ single crystals [3,23], whose *ZT* doubles in the (001) compared to along the (110), the *ZT* of n-type Bi_2_Te_3_ fibers is similar in the (001) orientation by Bridgman annealing compared to along the (110) orientation by whole annealing. In the meantime, during the bending test for estimating mechanical flexibility [9], the annealed fibers with a diameter *D* of 200 µm exhibit a minimum bending radius (*r*_min_) of about 1.8 cm and a maximum bending strain (*ε*_max_ = *D* / 2*r*_min_) of 0.56%.

## 4. Conclusions

In conclusion, high TE performance n-type Bi_2_Te_3_ fibers were prepared via rod-in-tube thermal drawing and Bridgman/whole annealing. The polycrystalline Bi_2_Te_3_ cores possess a polycrystalline orientation, and the *F* of (001) orientation factor in the Bridgman-annealed fiber cores is 0.92, and the *F* of (110) orientation factor in the whole-annealed fiber cores is 0.85. The (001) orientation increases the electrical conductivity and the thermal conductivity of the fiber cores more than the (110) orientation. Interestingly, both *ZT* of n-type Bi_2_Te_3_ fibers along (001) orientation and (110) orientation are approximately 1. Additionally, our future work will apply X-ray microfluorescence microscopy or tomography to study the composition of fibers in length and diameter. Finally, the Bridgman-annealed Bi_2_Te_3_ core shows an enhanced *ZT* = 1.05 at room temperature, and our future work will be on the TE device applications of these fibers. This proof-of-concept method of thermal drawing and annealing has potential in fiber-based TE applications.

## Figures and Tables

**Figure 1 nanomaterials-13-00326-f001:**
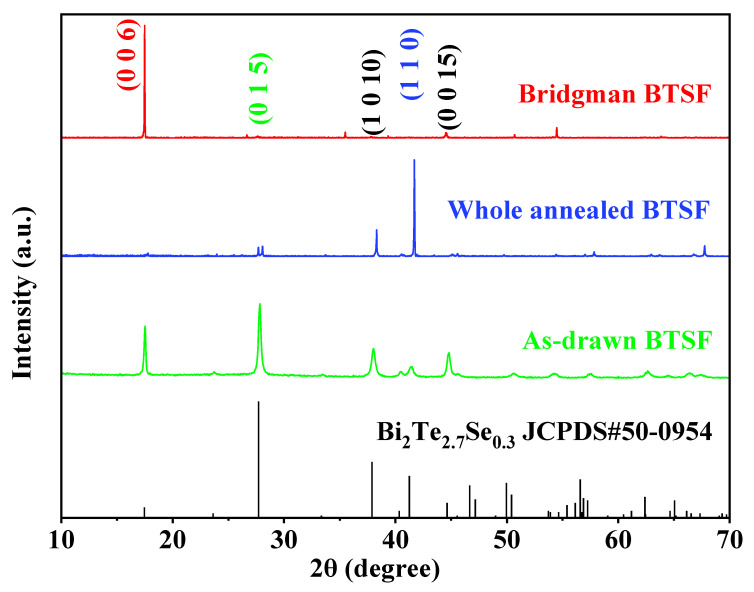
XRD patterns of the as-drawn and annealed BTSF.

**Figure 2 nanomaterials-13-00326-f002:**
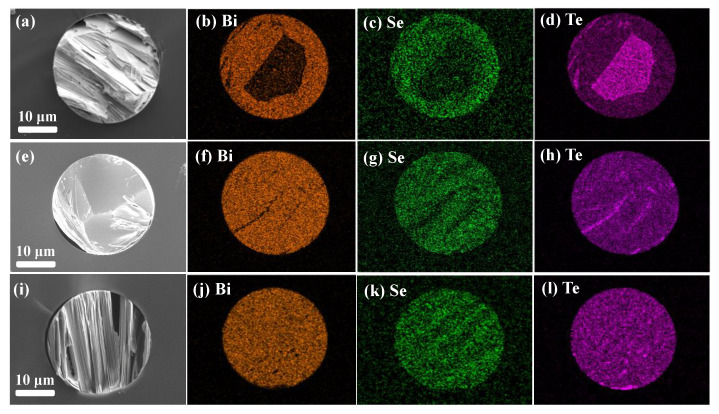
SEM cross-sectional images and EDS elemental mappings of (**a**–**d**) the as-drawn, (**e**–**h**) the whole-annealed, and (**i**–**l**) the Bridgman-annealed fibers.

**Figure 3 nanomaterials-13-00326-f003:**
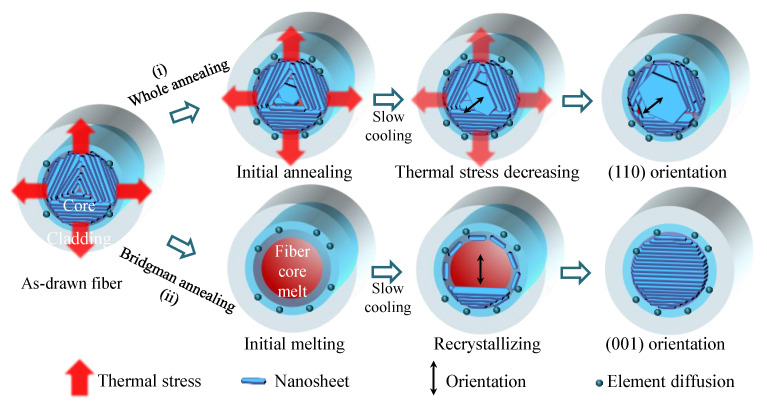
Two oriented crystal growth models of the n-type Bi_2_Te_3_ fiber core during whole annealing or Bridgman annealing.

**Figure 4 nanomaterials-13-00326-f004:**
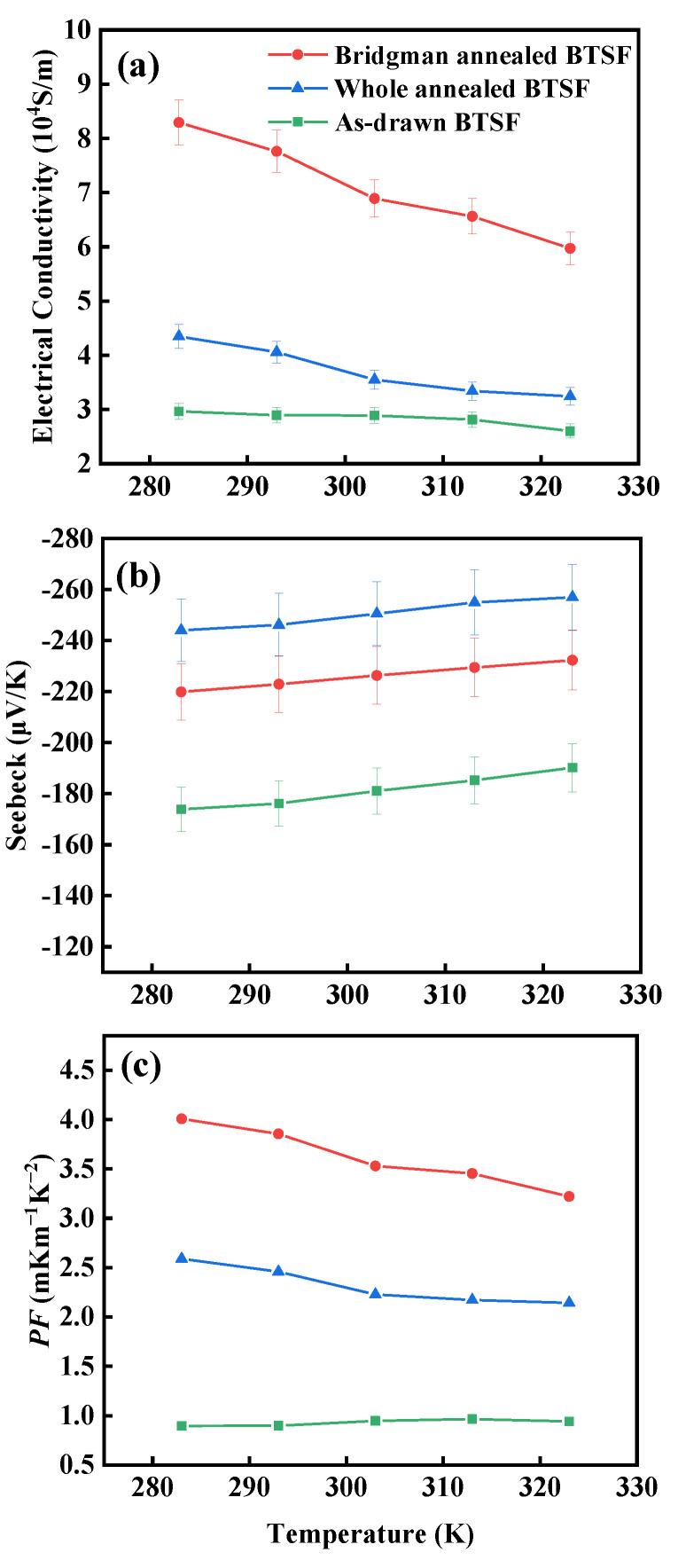
(**a**) The *σ*, (**b**) *S*, and (**c**) calculated *PF* of the as-drawn fibers and the annealed fibers at 10–50 °C.

**Table 1 nanomaterials-13-00326-t001:** *σ*, *S*, *κ*es, and calculated *ZT* of the n-type Bi_2_Te_3_-based fibers and Bridgman-method-growth crystals near room temperature.

Samples	Electrical Conductivity *σ* (S/cm)	Seebeck Coefficient *S* (μV/K)	Thermal Conductivity *κ* (W/mK)	*ZT*
Bridgman-annealed fibers	689 ± 33	−226 ± 11	1.01 ± 0.1	1.05
Whole-annealed fibers	355 ± 17	−251 ± 12	0.71 ± 0.07	0.95
As-drawn fibers	289 ± 14	−181 ± 9	0.59 ± 0.06	0.48
Bi_2_Te_2.5_Se_0.5_ core fibers [17]	180	−227	0.64	0.43
Bi_2_Se_3_ fibers [13]	763	−92	0.84	0.23
Bi_2_Se_3_ core fibers [15]	319	−150	1.25	0.18
Bridgman Bi-Te-Se crystals [25]	1064	−201	1.4	0.92

## Data Availability

The production data are available on request from the corresponding author.

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
