# Peer review of "High-Performance n-Type Bi_2_Te_3_ Thermoelectric Fibers with Oriented Crystal Nanosheets"

_nanomaterials, 2023, doi:10.3390/nano13020326_

Round 1
Reviewer 1 Report
The article is devoted to preparation and investigation of thermoelectric properties of Bi2Te3 fibers. As a thermoelectric material for Peltier microrefrigerators Bi2Te3 is well known. Bi2Te3 crystals are grown by the Bridgman method. In this case, in the production of thermoelectric elements, a group technology can be used, crystals are cut into small cubic elements, from which the thermoelectric elements are subsequently assembled.
But this article deals specifically with the production of Bi2Te3 fibers using three approaches that improve the thermoelectric properties of the fibers. In principle, the article is well written, but a number of questions remain about the content.
(1) (1) In the introduction it is necessary to provide detailed information about the application of thermoelectric fibers Bi2Te3. It is not obvious from the text of the article.
(2) It is necessary to compare the thermoelectric properties of fibers and Bi2Te3 crystals grown by the Bridgman method.
(3) From the article it seems not very technological to produce fibers, because in the final step the glass capillaries have to be dissolved in HF.
(4) And it is not at all clear how to use the fibers in the fabrication of thermoelectric elements.
And as a study of the properties of thermoelectric fibers in the article everything is presented correctly and interesting. Maybe it would be interesting to apply X-ray microfluorescence microscopy or tomography to study in detail the composition of fibers in length and diameter.
25 December 2022
Author Response
Responses to the Reviewers’ comments: Reviewer #1: The article is devoted to preparation and investigation of thermoelectric properties of Bi2Te3 fibers. As a thermoelectric material for Peltier microrefrigerators Bi2Te3 is well known. Bi2Te3 crystals are grown by the Bridgman method. In this case, in the production of thermoelectric elements, a group technology can be used, crystals are cut into small cubic elements, from which the thermoelectric elements are subsequently assembled. But this article deals specifically with the production of Bi2Te3 fibers using three approaches that improve the thermoelectric properties of the fibers. In principle, the article is well written, but a number of questions remain about the content. Response: We sincerely thank the reviewer for his/her recommendation that the article is well written. And the related answers to the questions are attached below. (1) In the introduction it is necessary to provide detailed information about the application of thermoelectric fibers Bi2Te3. It is not obvious from the text of the article. Response: The high-performance Bi2Te3 thermoelectric fibers could be used in the field of waste heat recycling on the curved surface (such as hot water tubes and vehicle tailpipes), Peltier-cooling and temperature-sensing textiles (such as face masks and sleeveless shirts), etc. Following the reviewer’s advice, we have added the application of thermoelectric fibers Bi2Te3 at the end of Introduction in the revised manuscript. (2) It is necessary to compare the thermoelectric properties of fibers and Bi2Te3 crystals grown by the Bridgman method. Response: Thanks for the reviewer’s constructive advice. The thermoelectric properties of n-type Bi2Te3 crystals grown by the Bridgman method are added at the end of Table 1 of the revised manuscript. The n-type Bi2Te3 crystals grown by the Bridgman method possess a ZT~0.92 [Ref.26: Journal of Materials Science, 2002, 37, 2893], which shows higher electrical and thermal conductivities but lower Seebeck coefficients than that of the Bridgman-annealed fibers. The sentence has been added in the last paragraph before the revised Table 1. (3) From the article it seems not very technological to produce fibers, because in the final step the glass capillaries have to be dissolved in HF. Response: Thanks for the reviewer’s good question. Indeed, the thermoelectric fibers with capillaries/claddings can be produced and applied without HF etching. The HF-etched fiber cores are mainly prepared for characterizing the crystallinity and electrical transport of the fiber cores. And the related explanation has been made clear at the end of Introduction and the beginning of 2.2Measurements in the revised manuscript. (4) And it is not at all clear how to use the fibers in the fabrication of thermoelectric elements. Response: Thanks for the reviewer’s comment about the fabrication of thermoelectric elements. To fabricate thermoelectric elements, such as the p-n fiber elements on thermoelectric cup generator or in thermoelectric cooling fabric [Ref.13: Nano Energy 2017, 41, 35], the n-type Bi2Te3 fibers with oriented crystal nanosheets in the core and the glass cladding protection should be connected by electroconductive paste electrically in series and thermally in parallel with the similar p-type Bi2Te3 thermoelectric fibers [Ref.6: Journal of Materiomics, 2020, 6, 467; Ref.9 Advanced Materials, 2022, 34, 2202942]. And the related explanation has been made clear at the end of Introduction in the revised manuscript. And as a study of the properties of thermoelectric fibers in the article everything is presented correctly and interesting. Maybe it would be interesting to apply X-ray microfluorescence microscopy or tomography to study in detail the composition of fibers in length and diameter. Response: We thank the reviewer very much for the valuable recommendation of X-ray microfluorescence microscopy or tomography to study the composition of fibers in length and diameter. The study will be our future work and the related description has been added to the Conclusion of the revised manuscript.Reviewer 2 Report
In the manuscript “High-performance n-type Bi2Te3 theroelectric fibers with oriented crystal nanosheets” the author has explored the use of Bi2Te3 for enhancing thermoelectric performance. There are a few issues that need to be addressed before accepting the manuscript for publication.
1 The language of the manuscript needs to be addressed for coherence. A lot of places are a little confusing to read. It would be better if the author makes it easier for reading.
2 Were the fibers kept under stress while performing the annealing (both) processes?
3 Line 107: The whole annealed fiber has a decrease in its orientation factor as compared to the as-drawn fiber, can the author comment on why the whole annealing process is inefficient for these kinds of fiber?
4 Figure 4 needs a better explanation of the obtained results. Figure 4c has a good explanation, it would be nice if the same is done for (a) and (b) as well. Explain the scientific merits instead of simply describing the results as it is. Why does the whole annealed sample have a higher Seebeck coefficient than Bridgman annealed samples while it is superior in every other factor?
5 I am curious to know the mechanical properties of the samples. Are they strong enough to be used for practical applications? Can the author include some results of a mechanical test?
6 The 001 orientation of the fibers lead to an increase in their properties if I understood correctly. It reported that the as-draw fibers has a higher orientation factor (F = 0.45) compared to the whole annealed samples (F=0.06). I am confused by the whole annealed samples still have better properties as reported in table 1. Why does this hold true only for Bridgman-annealed fibers?
Author Response
Responses to the Reviewers’ comments:
Reviewer #2:
In the manuscript “High-performance n-type Bi2Te3 thermoelectric fibers with oriented crystal nanosheets” the author has explored the use of Bi2Te3 for enhancing thermoelectric performance. There are a few issues that need to be addressed before accepting the manuscript for publication.
Response: We sincerely thank the reviewer for the recommendation of accepting the manuscript for publication after a few issues are addressed. And the related responses to the issues are attached below.
1 The language of the manuscript needs to be addressed for coherence. A lot of places are a little confusing to read. It would be better if the author makes it easier for reading.
Response: Thanks for the reviewer’s careful reading. We have rewritten the manuscript's language for coherence, such as (1) “Herein, high-performance n-type Bi2Te3 fibers were fabricated by optical-fiber-template thermal drawing (rod-in-tube method) , and subsequent annealing processes (whole annealing and Bridgman-type annealing) were used to enhance the crystallization of two kinds of opposite crystalline orientations” at the end of Introduction has been changed to “Herein, high-performance n-type Bi2Te3 fibers were fabricated by a rod-in-tube thermal drawing method, and subsequent annealing processes (whole annealing or Bridgman annealing) were used to enhance the crystallization of two kinds of opposite crystalline orientations”; (2) In Results and discussion, all the “(001) preferred orientation” or “c-plane (001) orientation” words have been changed to “(001) orientation”, all the “fiber core samples” words have been changed to “fiber cores” , and so on.
2 Were the fibers kept under stress while performing the annealing (both) processes?
Response: Thanks for the reviewer’s good question. The answer is no, and the fibers did not keep under external stress while performing the annealing processes. Indeed, the fiber cores were under tensile stress from cladding after the thermal drawing and quick cooling (>100°C/s) process as studied in our previous work [Ref.6: Journal of Materiomics, 2020, 6, 467]. And the related explanation has been made clear in the second sentence after Figure 3 of the revised manuscript.
3 Line 107: The whole annealed fiber has a decrease in its orientation factor as compared to the as-drawn fiber, can the author comment on why the whole annealing process is inefficient for these kinds of fiber?
Response: Thanks for the reviewer’s kind reminder. The related revision has been added to the end of the first paragraph after Figure 1 of the revised manuscript. The whole annealed fiber has a decrease in its (001) orientation factor (F(001)~0.06) but an increase in its (110) orientation factor (F(110)~0.85) as compared to the as-drawn fiber as shown in Figure 1. So the whole annealing process is inefficient to increase (001) orientation factor but efficient to increase (110) orientation factor for these kinds of fiber. The 565°C whole annealing should be a slow stress-relief process for the glass cladding with a 562°C glass-transition temperature since the as-drawn fiber cores were under tensile stress from the cladding as shown in Figure 3. Differently, the 645°C Bridgman annealing should be a recrystallization process for the 585°C melting temperature of Bi2Te3.
4 Figure 4 needs a better explanation of the obtained results. Figure 4c has a good explanation, it would be nice if the same is done for (a) and (b) as well. Explain the scientific merits instead of simply describing the results as it is. Why does the whole annealed sample have a higher Seebeck coefficient than Bridgman annealed samples while it is superior in every other factor?
Response: Thanks a lot for the reviewer’s constructive suggestion. Following the reviewer’s suggestion, we have added the related scientific discussion of electrical conductivities and Seebeck coefficients into the first paragraph after Figure 4 of the revised manuscript. The |Sw| of the whole-annealed fibers is bigger than |Sb| of the Bridgman-annealed fibers and |Sa| of the as-drawn fibers under a same temperature. This could be attributed to that the annealing process could decrease Te enrichment to increase |S|, and the oriented crystals along (110) should exhibit a little higher |S| than the oriented crystals along (001), whose isotropic behavior is following the result of the reported n-type Bi2T3 single crystals [Ref.23: Journal of Electronic Materials, 2010, 39, 1861] or films [Ref.24: Scientific Reports, 2016, 6, 1].
5 I am curious to know the mechanical properties of the samples. Are they strong enough to be used for practical applications? Can the author include some results of a mechanical test?
Response: We agree with the reviewer that mechanical properties need to be strong in practical applications. Following the reviewer’s comments, we have tested the bending radius of annealed fibers to estimate mechanical flexibility [Ref.17 Materials, 2022, 15, 5331]. The annealed fibers with a diameter (D) of 200 µm possess a minimum bending radius (rmin) of about 1.8 cm and a maximum bending strain (εmax=D/2rmin) of 0.56%. And the related revision has been added to the last sentence before Table 1 of the revised manuscript.
6 The 001 orientation of the fibers lead to an increase in their properties if I understood correctly. It reported that the as-draw fibers has a higher orientation factor (F=0.45) compared to the whole annealed samples (F=0.06). I am confused by the whole annealed samples still have better properties as reported in table 1. Why does this hold true only for Bridgman-annealed fibers?
Response: We sincerely thank the reviewer for his/her comment to improve our manuscript together. Indeed, different orientations in annealed fibers might lead to an increase in thermoelectric performance, since the annealed fibers possess a higher Seebeck coefficient than that of the as-drawn fibers because of reduced Te enrichment. The Bridgman-annealed fiber has an increase in its (001) orientation factor (F(001)~0.92) and the whole annealed fiber has a decrease in its (001) orientation factor (F(001)~0.06) but an increase in its (110) orientation factor (F(110)~0.85), as compared to the as-drawn fiber. And the related revision has been added to the end of the first paragraph after Figure 1 and the first paragraph after Figure 4 of the revised manuscript.
Round 2
Reviewer 1 Report
The authors provided a comprehensive response to the comments. The article may be published in present form.
Reviewer 2 Report
The author has done a good job addressing the issues and improved the manuscript's quality. The manuscript can be accepted in the present form after minor spell checks.